# Nanoarchitectonics of Electrically Activable Phosphonium Self-Assembled Monolayers to Efficiently Kill and Tackle Bacterial Infections on Demand

**DOI:** 10.3390/ijms23042183

**Published:** 2022-02-16

**Authors:** Serena Carrara, Florent Rouvier, Sanjana Auditto, Frédéric Brunel, Charlotte Jeanneau, Michel Camplo, Michelle Sergent, Imad About, Jean-Michel Bolla, Jean-Manuel Raimundo

**Affiliations:** 1Aix-Marseille Université, CNRS, CINAM, 13288 Marseille, France; serenella.carrara@gmail.com (S.C.); auditto@cinam.univ-mrs.fr (S.A.); fred.brunel@gmail.com (F.B.); michel.camplo@univ-amu.fr (M.C.); 2Aix-Marseille Université, INSERM, SSA, IRBA, MCT, 13005 Marseille, France; rouv.flo@hotmail.fr; 3Aix-Marseille Université, CNRS, ISM, Inst Movement Sci, 13385 Marseille, France; charlotte.janneau@univ-amu.fr (C.J.); imad.about@univ-amu.fr (I.A.); 4Aix-Marseille Université, CNRS, IRD, IMBE, 13397 Marseille, France; michelle.sergent@univ-amu.fr; 5Avignon Université, CNRS, IRD, IMBE, 13397 Marseille, France

**Keywords:** phosphoniums, self-assembled monolayers, biocidal effect, electroactivation, responsive surfaces

## Abstract

Prosthetic implants are widely used in dentistry and orthopedics and, as a result, infections can occur which cause their removal. Therefore, it is essential to propose methods of eradicating the bacteria that remain on the prosthesis during treatment. For this purpose, it is necessary to develop surfaces whose antibacterial activity can be controlled. Herein, we designed innovative and smart phosphonium self-assembled monolayer (SAM) interfaces that can be electrically activated on demand for controlling bacterial contaminations on solid surfaces. Upon electroactivation with a low potential (0.2 V for 60 min., conditions determined through a *DOE*), a successful stamping out of Gram-positive and Gram-negative bacterial strains was obtained with SAM-modified titanium surfaces, effectively killing 95% of *Staphylococcus aureus* and 90% *Klebsiella*
*pneumoniae*. More importantly, no toxicity towards eukaryotic cells was observed which further enhances the biocompatible character of these novel surfaces for further implementation.

## 1. Introduction

Nowadays, bacterial infections constitute a major threat to public health [1] and many experts are predicting a net increase in the mortality by 2050 caused by the rapid growth of multidrug resistant bacteria over the past decade [2,3]. To prevent this alarming scenario, it is urgent to identify new antibiotics and strategies to control bacterial proliferation. Although medical implants lead to a significant improvement in the patient’s well-being and healthcare, there are several drawbacks including surgical risks during the placement or removal, implant failure, and more specifically, infections by pathogenic microorganisms which are important to circumvent. These implant infections are a result of a bacterial biofilm formation [4] in which bacteria are embedded into a complex matrix rendering them more recalcitrant to conventional treatments. Implant surfaces are non-vascularized abiotic materials rendering the common strategies inappropriate and ineffective. This global issue of biofilm colonization on surfaces has driven the scientific community to develop an increasing number of novel materials possessing anti-biofilm features associated to biocidal properties [5]. The most predominant strategies to tackle the bacterial colonization are mainly based on the development of biofilm disrupting agents [6]. Other antibiofilm approaches based on the inhibition of bacterial adhesion or proliferation have also prompted strong interest [7]. Those include surfaces impregnated with antimicrobial and antiseptic hubs and cuffs [8], immunoprophylaxis [9], quorum sensing interference, impairment of staphylococci adhesion or biofilm accumulation [10], immunotherapy, enzymatic disruption, or removal of the biofilm [11], immunomodulation and use of nanoparticles to deliver anti-biofilm agents [12]. Despite those treatments, biofilms can re-grow with infection relapses and prolonged treatments may develop microbes with increased resistance to biocides with cumulative side effects of drugs that may have a substantial impact on the patient’s microbiota ecology.

So far, to circumvent these drawbacks, polymer-based antifouling, or antibacterial surfaces, acting either by contact effect or continuous release of bactericidal substances (antibiotics, metal ions, peptides), have been developed to tackle bacterial adhesion and kill bacteria [13]. Furthermore, strategies that delay or prevent the continuous release of active substances have been developed [14] from bacterial infection microenvironment-responsive surfaces (pH, bacterial enzymes (lipases, hyaluronidases), etc.) [15] or light [16]. Although some of them appear promising, these coatings often lead to a gradual leaching of the biocidal substances. Therefore, these coatings are inadequate for long-term uses as infections on prostheses can occur within a relatively long time after implantation [17].

Interestingly, electrical activation has been thought to be a possible trigger for such applications [18] since it has been efficiently applied to control the binding and switching of molecules and biomolecules onto surfaces [19]. In addition, in the late 1960s [20,21], critical work on the bioelectric effect demonstrated some interesting capabilities for eliminating biofilms from colonized surfaces [22]. However, this work has been underexploited until very recently but is still in its early stages [23,24]. Moreover, we have recently demonstrated that engineered phosphoniums can be used as effective antibacterial scaffolds against bacteria from the ESKAPE group [25,26,27].

Thus, in line with our previous work and based on the potential benefits of the bioelectric effect, we have developed novel electrically responsive surfaces possessing efficient biocidal properties made from the unique combination of phosphonium self-assembled monolayers onto a conductive surface such as titanium which is commonly used for implant manufacturing (Figure 1). The results emerging from this work are helpful and should be used to create innovative electro-responsive surfaces that are of great interest, for example, in the design and development of smart dressings that can allow both to monitor and eradicate microbial infections. The whole process could be controlled using mobile technologies with the possibility to have an electronic medical record and will constitute a personal point of care for patients. [28,29]

## 2. Results and Discussion

The synthesis of the target trialkylphosphonium bromide derivatives **3**–**5** is depicted in Figure 1 following previous published procedures [25,26,27]. Target compounds were readily prepared in two consecutive steps from commercially available dibromo derivatives. The first step involves a nucleophilic substitution reaction on the dibromoalkyl starting building blocks with potassium thioacetate in acetone to afford the monofunctionalized intermediates 1–2 in 65 and 74% yield, respectively (see SI) [30]. The latter were subsequently treated with the appropriate trialkylphosphines using an eco-friendly method under neat conditions using a microwave irradiation at 200W (130 °C) for 3 h leading to the compounds 3–5 in 98, 55 and 43% yields, respectively.

All new compounds have been characterized by conventional spectroscopic analytical methods. The phosphoniums were preselected because of their biocidal efficiencies against *Staphylococcus aureus* and other Gram-positive strains from the ESKAPE group in their planktonic form according to our previous works [25,26,27].

Ti and its alloys are widely used in orthopedic and orthodontic implants [31] but exhibit a lack of antibacterial effectiveness activities and implant-associated rejection issues occur in the case of long-term uses. Furthermore, the biocompatibility, both at the micro- and nanoscales of Ti alloys-based implants, depend strongly on their composition and surface roughness [32,33] that can be improved by either a physical or chemical surface modification process. To this aim, self-assembled monolayers (SAMs) have proven to be an efficient approach to coherently improve metallic surface properties including those of Ti. [34] Indeed, through careful engineering, SAMs act as dense, compact, and versatile layers that allow the modification and fine tuning of the surfaces with advanced and specific properties even on complex geometries.

The self-assembled monolayer formation onto the titanium disks was monitored by several methods including contact angle and SEM and compared with bare titanium surfaces. The contact angle of the functionalized surfaces decreased linearly with time compared to the bare surface due to the presence of the cationic phosphonium head groups suggesting the formation of a densely packed monolayer. This trait was also revealed by SEM micrographs demonstrating an increase in the surface smoothness along the time from bare titanium to the fully coated surface after 7 days (Figure 2a–c). Furthermore, EIS measurements were used to prove the SAM formation. A solid electrolyte (agarose 1%) in water was selected to conduct these experiments. Hence, the formation of the SAM is expected to gradually modify the level of charges and dipole at the interface leading to a modification of the capacitance and resistance (Figure 2d). The real impedance Z_re_ corresponding to the resistance at the SAM/agarose interface is obtained from the EIS spectra in a Nyquist plot. In this configuration, the electrode/solution interface is modeled as a simplified Randles equivalent circuit where the solution resistance and the double layer capacitance are in series. With time, the formation of the SAM steadily changes both the overall cell resistance and the capacitance since the molecular adsorption involves at least the displacement of water and ions from the interface. The resistance values were plotted against the time to generate an adsorption isotherm as depicted in Figure 2e.

SAM-modified titanium surfaces serve as models for dental implants, and their antibacterial properties have been evaluated with and without electrical activation. Gram positive *Staphylococcus aureus* and Gram-negative *Klebsiella pneumoniae* were chosen as species for the proof-of-principle tests and the antibacterial effect was determined according to the percentage of the bacterial survival after each experiment. As growth controls, a glass slit was used to replace the titanium disks. Electrical activation and experiments were performed using the 3D-imprinted device depicted in the Materials and Methods section and were carried out under the same conditions. An amount of 0.5 × 10^6^ Colony Forming Unit CFU (5 µL of a 10^8^ CFU/mL bacterial suspension) was deposited on top of the agarose surface prior to being sandwiched with the bare titanium or the SAM-modified titanium surface followed by securing the whole device to ensure watertightness. At the end of the experiments, the agarose and titanium disks were shaken in MH II media to additionally count the bacterial colonies in the suspensions. Several combinations of different parameters can be tested at the same time, for instance, the nature of molecular scaffold, the percentage of the SAM coverage, the type of electrochemical activation (CV/CA) and the duration (30, 60, 120 min.). Due to the large number of possible experiments, a Design of Experiment (DOE) has been set in order to obtain the best combination of parameters from a statistical plan based on a matrix calculation. The DOE offers a straightforward strategy to determine the best experimental choices that enhance the quality of the results. It helps in optimizing the parameters and gives a prompt statistical approach which is useful for an industrial application. For an overview, the dataset extrapolated from the matrix of DOE is reported in the Appendix A. All the combinations described by the DOE have been tested in triplicates to obtain a weighted value and a standard deviation at the end. Among them (see ESI for more details), the most efficient SAM which gave a great statistical meaning, and the best experimental action was compound 3 with a coverage of 100% by applying a voltage of +0.2 V for 60 min (Figure 3). Upon these conditions (+0.2 V, 60 min.), this functionalized surface, in contact with *S. aureus*, displayed a remarkable biocidal efficiency up to 95% compared to the untreated surface used as a reference. Encouraged by these results, experiments on Gram-negative bacteria have also been investigated by using the optimal conditions found for Gram-positive bacteria attributed to the DOE. Interestingly, up to 90% efficient biocidal effect is also observed for *K. pneumoniae*. The biocidal activities against Gram-positive and Gram-negative bacterial strains are encouraging and showcases the promising potential of an individual coating against a broad spectrum of bacteria.

On the contrary, when no electrical step is applied, a pre-treatment of the surface was optimized to observe a full inhibition of the biocidal effect, wherein the molecules were forced to conformationally bend by applying a potential of −0.4V for 5 min before being exposed to bacteria. In this way, the bacteria viability at the interface remains close to 100%, indicating a minimal/unhesitant effect of the molecule. This pre-treatment step notably shows that it is possible to inhibit almost completely the bacteria death observing only a small loss (Figure 3). However, if the latter step is not considered the SAM-modified Ti (3) surface has a considerable action toward the bacteria death, reaching a loss equal to 70% (Figure 4). In a previous study, we have already shown that phosphonium-based ionic liquids can kill the ESKAPE group of bacterial species with low minimal inhibitory concentration (MIC), including the *S. aureus* and *K. pneumoniae* strains tested here [27]. In this case, the compounds were solubilized, dispensed into the bacterial culture medium that contained 5 × 10 exp5 CFU/mL at a final concentration, and incubated for approximately 18 h, in accordance with international standard conditions for testing antimicrobial molecules (see ref [26] for details). In the present study, compound 3, an IL of the same structural basis, was self-assembled onto the titanium surface, which could strongly affect its effectiveness. Before evaluating the efficacy of SAM, the MIC of compound **3** was determined to be 2 μg/mL for *S. aureus* and 16 µg/mL for *K. pneumoniae*. In the experimental conditions for testing SAM, the incubation time did not exceed 60 min and the number of bacteria was about 5 × 10^5^ CFU/5 μL which was strongly different from MIC determination. However, a current of 0.2 V was applied during the whole incubation period and, although we did not achieve the same efficiency as in the assay in liquid, we still observed a strong decrease in CFU numbers. These results demonstrated that a clever combination of an effective molecule with a weak electric field can make a SAM surface capable of hindering bacterial colonization of at least *S. aureus* and *K. pneumoniae*.

In a second set of experiments, the different coatings have been tested regarding their safety towards eukaryotic cells. To conduct such experiments, a novel 3D-imprinted device has been designed and developed to perfectly fit on a 24-well culture plate containing a Boyden chamber with the same diameter of the titanium disks (Figure 5a). Within this configuration, the whole cell culture area is in contact with the SAM-modified titanium surfaces also avoiding a displacement of the titanium disk that could lead to the destruction of the parasitic cells hindering the real action of the modified surfaces. Furthermore, the 3D-imprinted device allows the precise placement of the electrodes in height and width to the nearest mm to ensure the reproducibility of the electrochemical tests.

Fibroblast cells were selected to evaluate the biocompatibility of the coatings with and without electrical activation and compared to a control. Without electroactivation, the cells’ viability is similar to the control, corroborating the non-cytotoxicity of the **3**-modified titanium surfaces under these conditions. More interestingly, upon the application of the potential i.e., 0.2 V over one hour the periodontal ligament cells remain unaffected demonstrating that the electroactivation conditions are ineffective in disrupting the fibroblast cells membrane (Figure 5b).

## 3. Materials and Methods

The reagents and materials are as follows. 1,12-dibromododecane, 1,6-dibromohexane, tri-n-hexylphosphine, tri-n-octylphosphine, potassium thioacetate were purchased and used as received from Alfa Aesar, TCI and Sigma-Aldrich. Column chromatography was performed using silica 60 M (0.04–0.063 mm) purchased from Macherey-Nagel. Solvents such as diethyl ether (Et_2_O), acetone, dichloromethane (DCM) were purchased from Sigma-Aldrich. Microwave irradiation experiments were conducted on a CEM Discover-SP apparatus (200 W) at 130 °C for 3 h. Agarose was purchased at Invitrogen (St. Quentin Fallavier, France). Bacteria culture materials were obtained from Dominique Dutscher (Brumath, France) and the culture media were obtained from Sigma-Aldrich (Saint-Quentin Fallavier, France). The following strains were used in this study: *Staphylococcus aureus* DSM 20231 and *Klebsiella pneumoniae* DSM 102040. The strains were purchase at the Leibniz Institute DSMZ-German Collection of Microorganisms and Cell Cultures GmbH. Cell culture materials and reagents were obtained from Dominique Dutscher (Brumath, France).

Human periodontal ligament cell culture: human primary periodontal ligament (hPDL) cells were prepared from immature third molars extracted for orthodontic reasons (male and female patients younger than 18 years) in compliance with French ethical legislation by the explant outgrowth method, as described previously [35]. These cells were grown in minimal essential medium (MEM) supplemented with 10% fetal bovine serum, 100 U/mL penicillin, 100 mg/mL streptomycin, and 0.25 mg/mL amphotericin B at 37 °C, 5% CO_2_ atmosphere [36].

Bacterial and cell cultures: rehydration of dried cultures was performed as recommended by the DSMZ. All microbial strains were stored at −80 °C in cryovial in Mueller–Hinton Broth II (MHII) supplemented with 30% (*v*/*v*) glycerol. Before testing, every week fresh cultures were initiated on MHII agar plate. Each day, an overnight culture was performed, then a culture expansion was performed before testing.

Evaluation of interface toxicity to hPDL cells: toxicity assays were performed in Boyden chambers; hPDL cells were plated in the upper chambers (25,000 cells in 100 μL) overnight. The Ti-SAM electrode was in contact with cells in the Boyden chamber, while the two Pt electrodes passed through the holes in the culture well. According to the device configuration, different conditions of electrostimulation were performed by varying the potential and duration. The medium was removed from the Boyden chamber and immediately replaced with 100 μL/well MTT solution (5 mg/mL). After incubation for 2 h at 37 °C, the supernatant was discarded, and the formed formazan crystals were solubilized with dimethyl sulfoxide (Sigma-Aldrich) (300 μL/well). A total of 100 μL of each solution was transferred on 96-well dishes. Then the absorbance of each well was determined using an automatic microplate spectrophotometer (Infinite 200; Tecan, Lyon, France) at 550 nm.

Physicochemical analysis: ^1^H, ^13^C and ^31^P NMR spectra were recorded on a JEOL ECS spectrometer at 400 MHz (^1^H), 100 MHz (^13^C) or 162 MHz (^31^P) at room temperature. NMR chemical shifts were given in ppm (¦Ä) relative to Me_4_Si with solvent resonances used as internal standards (CDCl_3_: 7.26 ppm for ^1^H and 77.2 ppm for ^13^C). ^31^P NMR spectra were given relative to external 80% H_3_PO_4_ standard. MS (ESI) analyses were performed on a SYNAPT G2 HDMS (Waters) spectrometer at the Spectropole of Aix-Marseille Université [37]. This spectrometer was equipped with an electrospray ionization source (ESI) and a time-of-flight (TOF) mass analyzer. The sample was ionized in the electrospray positive mode with a tension of 2.8 kV, the orifice tension was 50 V and the N_2_ flow rate was 100 L/h.

Contact angle measurements were performed at room temperature to access the hydrophobic/hydrophilic character of the modified substrates. Static contact angles (CA) were measured with an OCA 15 apparatus (DataPhysics) at room temperature using the sessile drop method and image analysis of the drop profile (SCA20 software). Deionized water droplet volume was 1 µL, and the contact angle was measured 10s after the drop was deposited onto the surface. Surface morphologies were conducted on untreated Ti and modified Ti using a scanning electron microscope (SEM JEOL JSM 6320F).

Physicochemical measurements in solution: electrochemical studies and non-faradic impedance measurements were performed on a VersaSTAT 4 potentiostat from Princeton Applied Research (Ametek scientific instruments, France). A three-electrode system based on a titanium disk (Ti) working electrode (diameter 10 mm, thickness 0.127 mm), a platinum (Pt) as a counter and a quasi-reference electrode were used. Tetrabutylammonium hexafluorophosphate (TBAPF6) served as a supporting electrolyte (0.1 M). Titanium disks (Ti) were cut from a 99.99% pure Ti sheet from Sigma-Aldrich and subsequently polished with diamond suspension (particle sizes 5 μm) from the ESCIL company. The freshly polished disks were subsequently rinsed and sonicated in distilled water, acetone and ethanol, then followed by blow-drying with nitrogen gas under an atmospheric condition. Then, the surfaces were exposed to a UV cleaning at 80 °C for 15 min prior to use. Lastly, they were deep coated in a 1 mM ethanolic solution of compounds 3–5 for 2 days, to get 50% coverage or 7 days to obtain 100% coverage. The coverage percentage was monitored by using electrochemical impedance spectroscopy. Cyclic voltammetry (CV) and chronoamperometry (CA) were used as standard methods to evaluate the charge and conformational changes effects on the bacterial death. CVs were performed at a scan rate of 0.1 V/s over a potential range of +0.2 V and −0.4 V for 30, 60 and 120 min while CA’s were performed at a different potential, +0.2 V, +0.5 V and +0.8 V for 30, 60 and 120 min. For negative control devices, a CA at −0.4 V for 5 min was preceding the bacterial test. Non-faradic impedance measurements were carried out at room temperature on the monolayers in the 10^5^ to 0.05 Hz frequency range using a modulation range of 25 mV in amplitude and a dc bias of zero.

Homemade imprinted device: a device that mimics a titanium/bacteria/periodontal ligament tissue implant interface was developed by using a 3D printer (Raise 3D Pro2) from Raise 3D Technologies, Inc. The device consists of a three-electrode electrochemical cell where the Ti surface (untreated or modified) is in contact with a 1% agarose gel that mimics the gum tissue and is also used as a solid electrolyte. The other two platinum electrodes, counter and quasi-reference, have been set on the side of the device and were selected due to their robustness and low interference over the antibacterial properties. A 5 μL bacterial suspension of *S. aureus* (strain DSM 20231) at a concentration of 10^8^ CFU/mL was deposited between the two elements (Figure 6).

**Synthesis of (12-(acetylthio)-dodecyl)-trihexylphosphonium bromide 3**: in an appropriate microwave vial 2 (238 mg; 0.74 mmol; 1 eq.) and trihexylphosphine (221 mg; 0.77 mmol; 1.05 eq.) were added under an inert atmosphere (glove box). Then the mixture was irradiated under microwaves (200W) at 130 °C for 3 h. The obtained viscous mixture was washed, triturated and centrifuged twice with Et_2_O, then the supernatant was removed, and the obtained residue was taken up into DCM and dried over a high vacuum. According to the NMR spectrum, the product was sufficiently pure to be used without further purification (449 mg; 0.73 mmol; 98% yield). ^1^H NMR (δ, 400 MHz; CDCl_3_): 2.84 (t, 2H, *J* = 7.2 Hz), 2.44 (br, 8H), 2.3 (s, 3H), 1.42–1.58 (br, 18H), 1.21–1.36 (br, 26H), 0.88 (t, 9H, *J* = 6.8 Hz). ^13^C NMR (δ, 101 MHz; CDCl_3_): 196.3, 31, 30.6, 29.5, 29.3, 29.2, 28.8, 22.4, 22, 19.7, 19.3, 14; 20 signals were obscured or overlapping. ^31^P NMR (δ, 162 MHz; CDCl_3_): 33.25. ESI MS (*m*/*z*): 529.4 (M^+^).

**Synthesis of (12-(acetylthio)-dodecyl)-trioctylphosphonium bromide 4**: in an appropriate microwave vial 2 (200 mg; 0.618 mmol; 1 eq) and trioctylphosphine (240 mg; 0.647 mmol; 1.05 eq) were added under inert atmosphere (glove box). The mixture was irradiated under microwave (200 W) at 130 °C for 3 h. The obtained viscous mixture was washed with Et_2_O and taken up in DCM and dried over high vacuum. The product was further purified via column chromatography using DCM: MeOH (40:1) affording (210 mg; 0.342 mmol; 55% yield) as a colorless viscous oil. ^1^H NMR (δ, 400 MHz; CDCl_3_): 2.84 (t, 2H, *J* = 7.3 Hz), 2.44 (br, 8H), 2.32 (s, 3H), 1.61 (br, 8H), 1.50–1.55 (m, 24H), 1.26 (br, 24H), 0.88 (t, 9H, *J* = 6.4 Hz). ^13^C NMR (δ, 101 MHz; CDCl_3_): 196.12, 31.66, 30.84, 30.69, 29.46, 29.27, 29.05, 28.94, 28.76, 22.57, 21.93, 21.89, 19.58, 19.11, 14.05. ^31^P NMR (δ, 162 MHz; CDCl_3_): 33.15. ESI MS (*m*/*z*): 613.5 (M^+^).

**Synthesis of (6-(acetylthio)-hexyl)-trioctylphosphonium bromide 5**: in an appropriate microwave vial 1 (200 mg; 0.836 mmol; 1 eq) and trioctylphosphine (309 mg; 0.833 mmol; 1.1 eq) were added under an inert atmosphere (glove box). The mixture was irradiated under microwaves (200 W) at 130 °C for 3 h. The obtained viscous mixture was washed with Et_2_O and taken up in DCM and dried over a high vacuum. The product was further purified via column chromatography using DCM: MeOH (40:1) to afford (190 mg; 0.358 mmol; 43% yield) as a colorless viscous oil. ^1^H NMR (δ, 400 MHz; CDCl_3_): 2.82 (t, 2H, *J* = 7.2 Hz), 2.46 (m, 8H), 2.31 (s, 3H), 1.71 (s, 2H), 1.45–1.56 (br, 18H), 1.31 (br, 24H), 0.87 (t, 9H, *J* = 6.3 Hz). ^13^C NMR (δ, 101 MHz; CDCl_3_): 196.02, 31.64, 30.82, 30.68, 30.63, 29.14, 28.93, 28.89, 28.69, 27.88, 22.55, 21.90, 21.86, 19.52, 19.05, 14.03. ^31^P NMR (δ, 162 MHz; CDCl_3_): 33.23. ESI MS (*m*/*z*): 529.5 (M^+^).

## 4. Conclusions

We report herein that the elaboration and characterization of a novel and innovative interface based on phosphonium self-assembled monolayers onto a conductive titanium surface can be used as an effective antibacterial coating under electrical activation. The new interface possesses remarkable biocidal properties when it is electro-activated at a low potential (0.2 V) over 1 hour. Under these conditions, that have been determined by the help of DOE, we successfully elaborated efficient antibacterial interfaces that unambiguously kill *Staphylococcus aureus* bacteria up to 95%. Interestingly, the novel interface has also proved to be very effective in eradicating Gram-negative bacteria such as *Klebsiella pneumoniae* by up to 90% under the same conditions. This unique combination of properties makes this innovative and novel interface a system of choice for the foreseen applications. The Design of Experiment (DOE) set up here limited considerably the number of experiments to allow us a first coverage of the study. We also considered the limitation of time to be compatible with a putative therapeutic application. In this context, while we may obtain an efficient decrease, there were still surviving bacteria, citing the need to upgrade and improve these surfaces to obtain more satisfying results. The goal must be to reach at least 99.99% of bacteria killing, and we are now considering several improvements; in particular, the development of SAM from new molecules which are currently under study. In addition, we consider that deeper studies would be necessary to better describe the mode of action of the SAMs under an electrical field.

## Data Availability

The data presented in this study are available on request from the corresponding author.

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
