# Peer review of "Nanoarchitectonics of Electrically Activable Phosphonium Self-Assembled Monolayers to Efficiently Kill and Tackle Bacterial Infections on Demand"

_ijms, 2022, doi:10.3390/ijms23042183_

Round 1

Reviewer 1 Report

This is a intresting experimental design, but these are some major defects as follows:

  1. What is antibacterial mechanism? It need be discussed in detail.
  2. How the broad-spectrum antibacterial activity of this self-assembled monolayer?
  3. As a implant, how to exert the potential in actual condition?
  4. Why negative polarization favor the growth of bacteria?
  5. What bacteria is shown in Fig.3(a)?
  6. With the increase of time applying the voltage, what is bacterial viability?

Reviewer 2 Report

- Add more current references (2020 – 2021).

- Add DOI for each article on references.

- Add more information about titanium alloys and his biocompatibility. In titanium alloys, the composition is very important, what alloying elements do it contain, for example V, Al cause side effects [1-4]. You can find more information in the cited works.

  1. Vizureanu, P.; Yamaguchi, S.; Le, P.; Baltatu, M. Biocompatibility Evaluation of New TiMoSi Alloys. Acta Phys. Pol. A 2020, 138, 283–286, doi:10.12693/aphyspola.138.283.
  2. Geetha, M.; Singh, A.K.; Asokamani, R.; Gogia, A.K. Ti based biomaterials, the ultimate choice for orthopaedic implants - A review. Sci. 2009, 54, 397-425. https://doi.org/10.1016/j.pmatsci.2008.06.004.
  3. Sidhu, S.S.; Singh, H.; Gepreel, M.A.-H. A review on alloy design, biological response, and strengthening of β-titanium alloys as biomaterials. Sci. Eng. C 2021, 121, 111661, doi:10.1016/j.msec.2020.111661.
  4. Baltatu, M.S.; Tugui, C.A.; Perju, M.C.; Benchea, M.; Spataru, M.C.; Sandu, A.V.; Vizureanu, P. Biocompatible titanium alloys used in medical applications. De Chim. 2019, 70, 1302–1306. https://doi.org/10.37358/RC.19.4.7114.

- Improve the introduction with data or future application for results obtained.

- The review of previous research should be much more detailed. Write specifically who did what and what results he came up with, in the considered research area.

- Which is the novelty of the work? Highlight this better

- What is the composition of the titanium alloys used?

- Complete the conclusions with the limitations of the proposed methodology. Also write future research. Put the conclusions at the end of the article.

- Generally, the quality of the writing could be improved.

Round 2

Reviewer 2 Report

The article has been significantly improved, it can be published.